# Circulating Factors as Potential Biomarkers of Cardiovascular Damage Progression Associated with Type 2 Diabetes

**DOI:** 10.3390/proteomes12040029

**Published:** 2024-10-11

**Authors:** Giovanni Sartore, Francesco Piarulli, Eugenio Ragazzi, Alice Mallia, Stefania Ghilardi, Massimo Carollo, Annunziata Lapolla, Cristina Banfi

**Affiliations:** 1Department of Medicine-DIMED, University of Padova, 35122 Padova, Italy; g.sartore@unipd.it (G.S.); francesco.piarulli@unipd.it (F.P.); drmassimocarollo@gmail.com (M.C.); annunziata.lapolla@unipd.it (A.L.); 2Studium Patavinum, University of Padova, 35122 Padova, Italy; 3Unit of Functional Proteomics, Metabolomics, and Network Analysis, Centro Cardiologico Monzino IRCCS, 20138 Milan, Italy; alice.mallia@cardiologicomonzino.it (A.M.); stefania.ghilardi@cardiologicomonzino.it (S.G.); cristina.banfi@cardiologicomonzino.it (C.B.)

**Keywords:** targeted proteomics, type 2 diabetes, cardiovascular disease

## Abstract

*Background*: Diabetes, particularly type 2 diabetes (T2D), is linked with an increased risk of developing coronary heart disease (CHD). The present study aimed to evaluate potential circulating biomarkers of CHD by adopting a targeted proteomic approach based on proximity extension assays (PEA). *Methods*: The study was based on 30 patients with both T2D and CHD (group DC), 30 patients with T2D without CHD (group DN) and 29 patients without diabetes but with a diagnosis of CHD (group NC). Plasma samples were analyzed using PEA, with an Olink Target 96 cardiometabolic panel expressed as normalized protein expression (NPX) units. *Results*: Lysosomal Pro-X carboxypeptidase (PRCP), Liver carboxylesterase 1 (CES1), Complement C2 (C2), and Intercellular adhesion molecule 3 (ICAM3) were lower in the DC and NC groups compared with the DN groups. Lithostathine-1-alpha (REG1A) and Immunoglobulin lambda constant 2 (IGLC2) were found higher in the DC group compared to DN and NC groups. ROC analysis suggested a significant ability of the six proteins to distinguish among the three groups (whole model test *p* < 0.0001, AUC 0.83–0.88), with a satisfactory discriminating performance in terms of sensitivity (77–90%) and specificity (70–90%). A possible role of IGLC2, PRCP, and REG1A in indicating kidney impairment was found, with a sensitivity of 92% and specificity of 83%. *Conclusions*: The identified panel of six plasma proteins, using a targeted proteomic approach, provided evidence that these parameters could be considered in the chronic evolution of T2D and its complications.

## 1. Introduction

Diabetes, particularly type 2 diabetes (T2D), is linked with a 2–4 times higher chance of developing coronary heart disease (CHD) [1], which is the primary cause of illness and death in developed nations. T2D is a condition that can negatively impact the prognosis of patients who have experienced acute myocardial infarction (MI), congestive heart failure, and coronary revascularization [2,3,4]. This is because T2D patients have an increased onset of atherosclerosis and atherothrombosis, which can be attributed to several factors, including endothelial dysfunction, dyslipidemia, chronic hyperglycemia, and insulin resistance. Furthermore, the prolonged duration of diabetes (>10–12 years) in men increases CHD mortality at a rate similar to that of men without diabetes but with a history of MI [5].

Identifying specific protein markers that can recognize different levels of disease severity or the progression of T2D associated with CHD is a relevant issue. Proteomic “signatures” made of multiple disease-relevant proteins could improve the management of complex diseases [6,7,8]. A multimarker approach assessing several biomarkers simultaneously can provide comprehensive information about the patient’s condition and improve personalized care by identifying individuals at high risk of disease progression. Through the utilization of a targeted mass spectrometry approach, we have recently established that 13 specific circulating proteins have the potential to serve as biomarkers for the progression of cardiovascular damage associated with T2D. Notably, this methodology has yielded exceptional classification outcomes with regard to sensitivity and specificity [9].

Herein, we extend our knowledge of potential circulating biomarkers by adopting a targeted proteomic approach based on proximity extension assays (PEA), which offers the advantage of increasing the plasma proteome coverage by merging a quantitative real-time polymerase chain reaction (qPCR) with multiplex immunoassays [10]. Indeed, proteins are more challenging to measure than DNA, and current proteomics technologies struggle with specificity, sensitivity, throughput, and dynamic range. The wide dynamic range of protein concentration in plasma presents a significant challenge in analyzing the plasma proteome. PEA is an advanced technology specifically developed for analyzing secreted proteins in serum and blood plasma. This groundbreaking technology has unequivocally demonstrated unparalleled specificity and sensitivity (sub-pg/mL), enabling the performance of high multiplex assays with extensive coverage across an extensive dynamic range (~9 log) while minimizing sample usage.

## 2. Materials and Methods

### 2.1. Patient Recruitment

The present investigation, which has the characteristics of a cross-sectional observational study, was conducted on 30 patients affected by both T2D and CHD (group DC), 30 patients with T2D without CHD (group DN), and 29 patients not affected by diabetes but with a diagnosis of CHD (group NC), attending the Center of Diabetology and Dietetics of Ulss 6 Euganea, Padova (Italy).

Patients with T2D (DC and DN groups) followed a Mediterranean-style diet (isocaloric) and were treated with individualized hypoglycemic therapy, consisting mainly of metformin. CHD was defined according to standard criteria and anamnestic data available from electronic records. Ninety percent of the NC group patients were assuming anti-hypertensive drugs. DC and NC patients were assuming low-dose aspirin (70% and 57%, respectively) and statins (79% and 60%, respectively). No differences in smoking habits were detected among the groups of patients. The study was conducted in the pre-COVID period. All patients were in good health without signs of heart failure (no dyspnea on exertion or rest, no declivous oedema; NYHA class 0/1) and without acute respiratory diseases. Patients with chronic obstructive pulmonary disease, asthma, a history and/or documentation of pulmonary embolism and/or pulmonary hypertension, as well as patients with severe cerebrovascular, renal, and hepatic and hematologic diseases were excluded from the study. The complete details about enrolled patients have been previously published [9,11]. The research was carried out in compliance with the guidelines of the Declaration of Helsinki and received approval by the Ethics Committee for Clinical Trials in the Province of Padova, reference study No. 97610. Informed written consent was obtained from all participants.

### 2.2. Sample Preparation

Plasma was obtained by centrifugation (1500× *g* for 15 min, 4 °C) from blood samples collected in citrate tubes (0.129 mmol/L). The plasma was subsequently divided into aliquots and stored at 80 °C until analysis. 

### 2.3. Circulating Proteome Profiling and Analysis 

The plasma samples were analyzed using the proximity extension assay (PEA) with Olink Target 96 cardiometabolic panel (Olink Proteomics, Uppsala, Sweden). In this technique, target proteins specifically bind to double oligonucleotide-labeled antibody probes. Subsequently, microfluidic real-time PCR amplifies the oligonucleotide sequence for quantitative DNA sequence detection. Quality control and normalization were applied to the threshold cycle (Ct) data obtained from internal and external controls. The list of proteins analyzed with respective acronym and Uniprot code is shown in Appendix A. Protein levels were quantified on a relative scale and expressed as normalized protein expression (NPX) units in log_2_ scale. Higher NPX values indicate higher protein concentrations. 

### 2.4. Statistical Analysis

The statistical analysis was conducted with JMP^®^ Pro 17 software (SAS Institute Inc., Cary, NC, USA) and MetaboAnalyst 5.0 [12]. NPX data were evaluated using the non-parametric multivariate method of principal component analysis (PCA) [13] and cluster analysis technique [12]. Analysis of data was obtained also with a parametric approach, where differences in protein plasma content across the patients’ groups were evaluated with one-way analysis of variance (ANOVA) followed by post hoc test with Fisher’s least significant difference (LSD) and Tukey’s HSD methods. Correlations between proteins were determined with Pearson’s *r* correlation coefficient. A *p*-value < 0.05 was considered as statistically significant. A receiver operating characteristic (ROC) analysis was carried out in order to predict the ability of the identified protein panel to classify the three groups of patients. Area under the ROC curve (AUC) was used to indicate the goodness of fit for the logistic model; significance was assessed by likelihood ratio chi-square test. Sensitivity and specificity were determined at optimal cut-off points.

## 3. Results

The clinical and metabolic data of the subjects are reported in Appendix A. Full information of the patients’ characteristics is available in a previously published study [11]. Significant differences between groups were observed for BMI, fasting plasma glucose, HbA1c, in particular for T2D patients (DN and DC groups) with respect to non-diabetic patients with CHD (NC group) (Appendix A). Diabetes duration in DC patients was significantly longer than DN patients (10.9 ± 7.6 y vs. 1.9 ± 0.9 y, *p* < 0.001).

Plasma samples from the three groups of subjects were analyzed by a targeted proteomic approach based on the PEA principle in order to detect relative changes in protein levels according to the group belonging. Appendix A shows the distribution profile of the investigated panel of 92 proteins as obtained in the overall subjects; Appendix A presents the protein data divided according to the three groups of patients. Differences in the proteomic profile among patient groups were evaluated using one-way ANOVA, revealing six significantly different proteins (Table 1 and Figure 1). A post hoc analysis with pairwise comparisons confirmed the differences between groups (Table 1). In particular, Lysosomal Pro-X carboxypeptidase (PRCP), Liver carboxylesterase 1 (CES1), Complement C2 (C2), and Intercellular adhesion molecule 3 (ICAM3) were lower in the DC and NC groups compared with the DN groups. Lithostathine-1-alpha (REG1A) and Immunoglobulin lambda constant 2 (IGLC2) were higher in the DC group than in DN and NC groups (Figure 1). 

To evaluate the combined discriminatory ability of all the 92 measured proteins across the three groups of patients, multivariate approaches were employed. However, an unsupervised analysis using PCA did not reveal any significant differentiation between the groups (Appendix A). Indeed, no clear distinction of the three groups appeared, since the scores plot presented overlapping clusters. Cluster analysis also conducted using all the 92 measured proteins did not reveal a clear separation of the three groups of subjects, as indicated by the heatmap of Appendix A. 

Statistical analysis was then focused on the six significant proteins, which revealed some common features in their behavior. The biplot from a PCA conducted on this restricted panel of proteins (Figure 2) permitted the identification of a very similar profile for IGLC2 and REG1A, demonstrated by the small angle formed between the vectors. The remaining proteins displayed a more dispersed range of distribution. To complete the view, also a linear correlation was evaluated between the 6 proteins (Figure 3; Appendix A shows statistical details), confirming significant relationships between almost all the proteins.

To evaluate the possible role of the six significant proteins in distinguishing among the three groups of subjects, ROC curves were obtained following a nominal logistic regression. Appendix A shows the result for the overall model, which suggests a significant distinction among the three groups (whole model test *p* < 0.0001). Figure 4 presents the ROC analysis directed at specific comparisons between groups. The model allows a satisfactory discriminating performance in terms of sensitivity and specificity (Table 2). The parameters of ROC analysis obtained for each single protein is presented in Appendix A; the performance of the overall model including the 6 proteins is better than that of each protein taken alone.

The role of the six identified plasma proteins was also evaluated in discriminating the kidney function. Taking the overall cohort of patients, the presence of a renal function impairment was considered for eGFR < 60 mL/min. A logistic regression with following ROC analysis (Appendix A) revealed a possible role of IGLC2, PRCP and to a lesser extent of REG1A in indicating kidney impairment, with a sensitivity of 92% and specificity of 83% (Appendix A). Appendix A also presents the estimated parameters from ROC analysis for each separate protein, confirming the predictive role on the presence of a kidney impaired function.

## 4. Discussion

The present investigation, performed by a targeted proteomic approach based on the PEA technology, revealed a panel of six plasma proteins which could represent a potential predictive tool for the development of diabetes-related cardiovascular complications and indicates possible applications of plasma proteome within precision medicine. The six proteins that resulted significantly different in our study were able to differentiate among subjects with diabetes, with or without CHD, and subjects with CHD but without diabetes, respectively. The multiple statistical approaches performed on the proteomic analytical data permitted to show a satisfactory ROC curve performance with good levels of sensitivity and specificity in order to distinguish between patients with diabetes, and patients with or without evolution to CHD, respectively.

The six proteins that emerged from the multivariate statistical analysis have some roles already demonstrated in the pathogenetic history of diabetes, and therefore can be here discussed.

Lysosomal Pro-X carboxypeptidase (PRCP) is a serine protease that acts on peptides in three major pathways: the pro-opiomelanocortin (POMC) system, the renin-angiotensin system (RAS), and the kallikrein-kinin system (KKS). PRCP regulates the anorexigenic properties of α-Melanocyte-stimulating hormone (α-MSH1-13) within the POMC system [14]. Mice lacking PRCP show increased levels of α-MSH in the hypothalamus and improved glucose metabolism on a high-fat diet compared to wild-type mice [14]. PRCP is active in the RAS and the KKS in the cardiovascular system. It is involved in vascular and cardiovascular homeostasis, tone regulation, and inflammation response. It converts kallicrein to bradykinin 1–9, which regulates the release of nitric oxide and prostaglandins, leading to vasodilation and inflammation [15]. Serum PRCP activity has been found to be higher in T2D in obese men [16], in comparison to lean and obese non-diabetic men, and a positive correlation was observed with glycemic control, and related to immune cells from visceral adipose tissue. In a study on ZDF rats fed a high-fat diet (HFD), Tabrizian et al. [17] found altered PRCP expression, which was linked to severe hyperglycemia and nephropathy. The data from the current study show that individuals in the DN group, who have recently developed T2D but do not have CHD, have higher levels of a PRCP in their plasma compared to individuals in the DC and NC groups. The higher level of PRCP found in plasma of DN subjects, who have a short diabetes duration history, may suggest an ongoing dysregulation in the inflammatory status during the early diabetes phases, and therefore could represent an additional prognostic factor to be evaluated in the follow-up of the disease. In our study, however, no correlation was found between PRCP in blood and FPG in any of the three patients’ groups (see Appendix A); only a marginally significant negative correlation was found in DN group with BMI, in contrast with the findings reported by De Hert et al. [16] who instead found a positive correlation with obesity. Therefore, the exact role of and pathogenesis linked to PRCP needs further investigation in relation to diabetes history.

Carboxylesterase 1 (CES1), an enzyme involved in lipid metabolism that hydrolyses triglycerides and cholesterol esters, has been demonstrated to possess a role in glucose homeostasis. In a mouse model, Xu et al. [18] have shown that hepatic CES1 overexpression is able to decrease plasma glucose and improve insulin sensitivity. Moreover, a decreased CES1 expression has been reported in a rat model of T2D [19]. The higher plasma concentration of CES1 observed in our DN group, in comparison to the other groups, suggests that in the early stages of diabetes, this protein could possess a protective role in the disease evolution that ceases with disease progression and proceeds toward cardiovascular complications.

Comparing the three groups of subjects, the plasma levels of C2 and ICAM3 showed a similar trend to those of PRCP and CES1. Studies have revealed that high levels of complement C2 in the blood plasma are linked with an increased risk of type 2 diabetes [20,21]. These findings support the existing knowledge about the role of the complement system in inflammation and how it adversely affects metabolism [20,21]. Shim et al. [22] have reported that activation of complement system is strictly linked to the initiation and progression of metabolic disorders, such as obesity, insulin resistance and diabetes. ICAM3 facilitates intercellular adhesion among leucocytes. This protein is expressed on resting antigen-presenting cells, such as dendritic cells, and plays a crucial role in the initial interactions between dendritic cells and T cells that lead to T cell activation [23]. 

The profiles of the four above discussed proteins exhibit similarities, as shown in Figure 1 and by the correlations (PCA and linear correlation: Figure 2 and Figure 3). It is therefore reasonable to consider them as a cluster that may deserve further investigation regarding their functional and pathophysiological interconnections.

The last two proteins identified, namely REG1A and IGLC2, display a similar profile in the three groups of patients, but differ if compared to the cluster of the four previously described proteins. 

REG1A, also known as lithostatine, belongs to a family of secreted proteins characterized by the presence of a C-type lectin domain. Its expression is observed across various organs and is implicated in vital cellular processes within the digestive system, including proliferation, differentiation, inflammation, and carcinogenesis [24]. The expression of REG1A has been demonstrated to be significantly increased in blood samples of patients affected by diabetic kidney disease (DKD) [25] and correlated with urinary albumin/creatinine ratio, suggesting a role as a biomarker for the risk of developing DKD. The present data agree with this observation, since our DC patients, affected by both T2D and CHD, have the highest plasma level of the protein REG1A. Therefore, a possible link of the protein with the evolution of diabetic kidney disease could be hypothesized. 

IGCL2, the immunoglobulin lambda constant 2, located in blood microparticles and extracellular exosome, is believed to influence antigen binding activity and immunoglobulin receptor binding activity. It is involved in several immune-related processes, including phagocytosis [26]. Osuna-Martinez et al. [27] reported that the IGLC2 gene is upregulated in samples of kidney tubules from patients with diabetic kidney disease (DKD). It is of interest to compare the higher level found in our cohort of DC patients, in comparison to the other two groups, suggesting a possible involvement in kidney disease evolution. Also of interest is the present finding related to the predictive role of IGLC2 and REG1A, with the contribution of PRCP, for the presence of impaired kidney function, as demonstrated by ROC analysis, confirming the relevance of the above-mentioned proteins as a possible marker of renovascular damage. The overlap in plasma concentrations of both REG1A and IGLC2 in patients affected by diabetes with (DC) and without CHD (DN) (box plot of Figure 1; close vector angle in PCA plot of Figure 2) supports a role of these plasma proteins throughout the time course evolution of diabetic cardiorenal disease. At present, no other links to metabolic or cardiovascular disease have been reported in the literature, and therefore our finding may be of interest as a possible predictor of DKD evolution, although the exact pathogenetic role remains to be elucidated.

The overall results of the present investigation suggest that the six identified plasma proteins are involved at various levels in patients affected by diabetes, with (DC group) or without CHD (DN group), but not in subjects affected by CHD only (NC group). A schematic representation of the hypothetical pathophysiological mechanisms is illustrated in Figure 5. In particular, considering the glucometabolic aspect, PRCP and CES1, which resulted here fairly well correlated (Figure 3) also according to PCA analysis (Figure 2), have been found linked to glycemic control (PRCP) [16] and insulin sensitivity (CES1) [18,19]. At the inflammatory level, since diabetes is considered a typical proinflammatory state accompanied by increased cellular inflammation [28], the close correlation found between PRCP and ICAM3 plasma levels (documented both by linear correlation in Figure 3 and by the tight vector angle in PCA plot of Figure 2) supports their role in endothelial dysfunction induced by glucotoxicity. In addition to glucotoxicity, insulin resistance presents a close link with endothelial dysfunction [29]. Insulin resistance is represented in this case by C2 [22], whose activation has been reported in initiation and progression of glucometabolic disorders; in this respect, the correlation detected here between C2 and ICAM3 supports the relevance of a dysfunction at endothelial level, which may lead to overt atherosclerosis even before the diagnosis of diabetes [30].

It can be of interest to make a comparison between the proteins found in the present study and those identified in our previous investigation [9] conducted on the same subjects using multiplexed MRM-based proteomics. The whole protein panel investigated in the two studies mainly did not coincide, with only about 10% proteins in common, so making the two investigations complementary. Although the different proteomic techniques cannot permit a direct quantitative comparison of data, the relative behavior of proteins in common was similar. By means of a multivariate approach (PCA) it is possible to observe that the overall pattern of the significant proteins indicates, besides absence of any overlapping, the complementary profile of the proteome (Appendix A), with a quite diversified distribution among the different proteins, presenting various degrees of affinities, as suggested by the dispersion of the loadings. In particular, since the vectors of the group of proteins ICAM3/PRCP/CES1/C2 identified in the present study show an angle near 90° in comparison to others, a low correlation with the other proteins is suggested. Although the comparison may be affected by the different method of analysis, it is possible to hypothesize that the whole pattern of proteins may serve to increase the number of potentially useful markers for cardiovascular complications of diabetes.

The present investigation has limitations. Since the protocol is related to our previous study [11], the data are based on a relatively small number of participants, and as a preliminary evaluation, it did not rely on a specific power analysis. A further limitation regards the absence of a healthy control group, that was not considered, since the aim of the present investigation was a comparison of the levels of plasma proteins in pathological conditions involving CHD, a complication typical in the history of T2D. Due to the fact that the present research is a cross-sectional observational study, we could not evaluate every lifestyle factor or comorbidity, so omitting possible residual confounding factors. It should be pointed out that T2D patients (DC and DN groups) were assuming anti-diabetic drugs, differently from NC group patients, and therefore the potential interaction of these drugs with the investigated protein levels remains to be elucidated. An additional limitation, due to the observational nature of the study, regards the difficulty to provide a causal inference for the identified proteins. However, the significance obtained with the present data suggests a role of the identified plasma proteins in the cardiovascular complications of patients with diabetes. The findings deserve confirmation with a larger sample size and with a more specific experimental protocol.

The identified protein species in plasma of patients with T2D suggest that attention should be given to the study of the proteome complexity also in this specific clinical diabetology field. Future research could possibly investigate, within the identified targets in the plasma proteome complexity, also the presence of proteoforms, that could help in distinguishing protein variants more specifically to be considered in the clinical history of T2D and its complications.

## 5. Conclusions

In conclusion, the present study, which identified a panel of six plasma proteins using a targeted proteomic approach, provides evidence that these parameters, in addition to the conventional risk factors and single disease biomarkers, could be considered in the chronic evolution of T2D and its complications. The identified plasma proteins are associated with the cardiovascular risk profile of T2D, aligning with our previous investigation on multiple circulating proteins which suggested possible biomarkers for cardiovascular damage progression in T2D [9]. The exact role of these proteins detected in the present study remains to be confirmed in wider scale future investigations.

## Figures and Tables

**Figure 1 proteomes-12-00029-f001:**
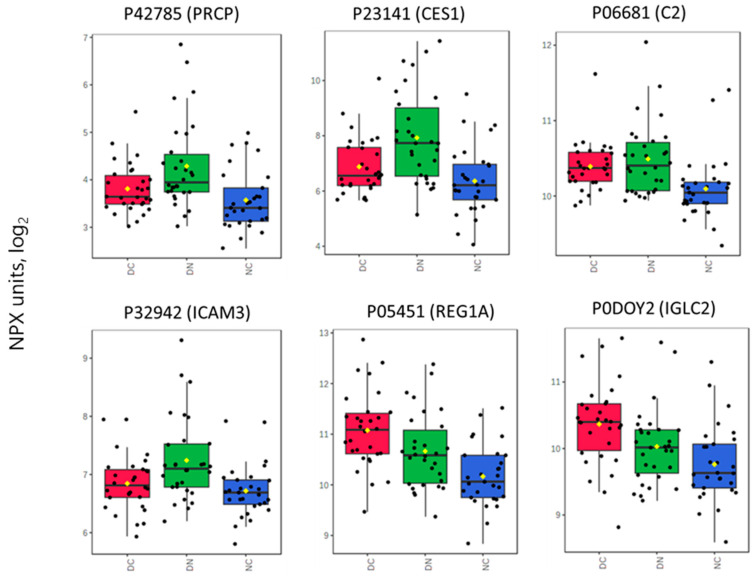
Box plots showing the distribution of the Normalized Protein eXpression (NPX) Units for proteins which significantly differ among groups. See Table 1 also for specifications and paired comparisons. Each box extends from the first quartile to the third quartile (interquartile range, IQR), while the horizontal line within each box represents the median. Whiskers extend to 1.5 × IQR. Yellow lozenge represents the mean.

**Figure 2 proteomes-12-00029-f002:**
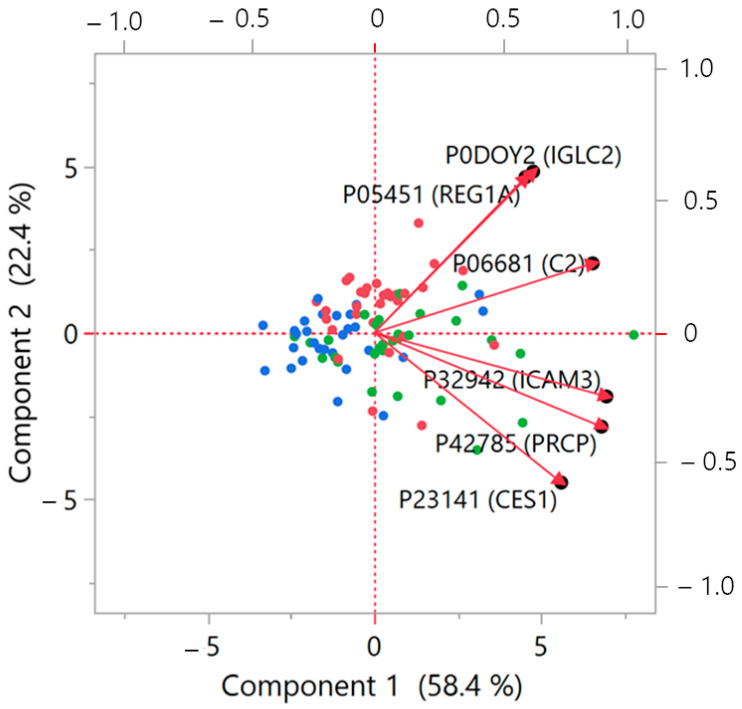
Biplot from PCA analysis on the six identified significant proteins. When vectors show a small angle, then the corresponding variables are positively correlated, while when the vectors are at 90°, they are not likely correlated. The biplot illustrates in a bidimensional space the multivariate distribution of the proteins, represented as vectors, together the points relative to the subjects investigated. Red dots: DC group; green: DN group; blue: NC group. The left and bottom axes show principal component scores; the top and right axes indicate the loadings. Further details about the structure of the PCA biplot can be found in [13].

**Figure 3 proteomes-12-00029-f003:**
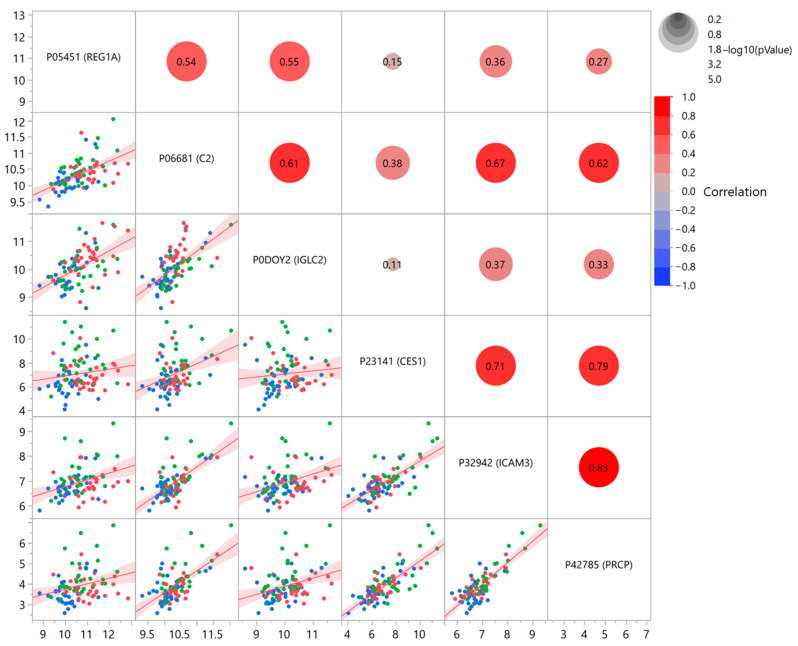
Linear correlation between the six identified significant proteins. Axes represent NPX units. The scatterplot shows all the pairwise comparisons, with the corresponding correlation coefficient *r*, represented also with a colored circle proportional to the degree of correlation. Red dots: DC group; green: DN group; blue: NC group.

**Figure 4 proteomes-12-00029-f004:**
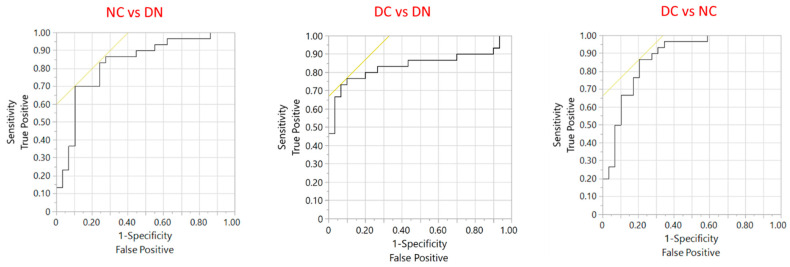
ROC analysis of the performance of the six identified proteins to distinguish between the three groups of patients. The black line is the ROC curve plot while the yellow line represents the tangent to the threshold point that maximizes the sum of sensitivity and specificity.

**Figure 5 proteomes-12-00029-f005:**
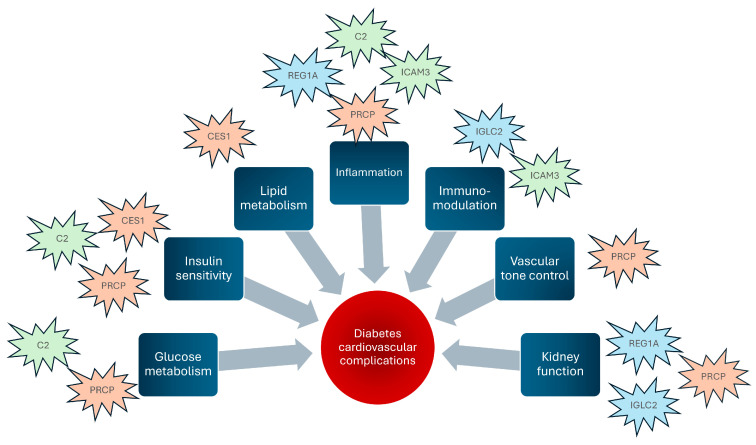
Schematic representation of how the identified proteins can act as pathophysiological factors leading to cardiovascular complications of diabetes. The color assigned to the proteins indicates similar behavior of the plasma profile in our subjects.

**Table 1 proteomes-12-00029-t001:** Proteins found to differ significantly (ANOVA) among the three patients’ groups. Post hoc analysis (Fisher’s LSD and Tukey’s HSD test) shows the pairwise comparisons which significantly differ between groups. False discovery rate (FDR) is also presented.

Protein UNIPROT Code	Protein Name (and Abbreviation)	ANOVA F Value	ANOVA *p*	−log_10_(*p*)	FDR	Fisher’s LSD Paired Comparisons	Tukey’s HSD Paired Comparisons
P05451	Lithostathine-1-alpha (REG1A)	12.229	2.12 × 10^−5^	4.6741	0.002	DC-DN; DC-NC; DN-NC	NC-DC; NC-DN
P23141	Liver carboxylesterase 1 (CES1)	10.625	7.52 × 10^−5^	4.1236	0.003	DN-DC; DN-NC	DN-DC; NC-DN
P0DOY2	Immunoglobulin lambda constant 2 (IGLC2)	7.7751	0.000787	3.1038	0.020	DC-DN; DC-NC	NC-DC
P42785	Lysosomal Pro-X carboxypeptidase (PRCP)	7.6633	0.000866	3.0627	0.020	DN-DC; DN-NC	DN-DC; NC-DN
P06681	Complement (C2)	7.2305	0.001252	2.9025	0.023	DC-NC; DN-NC	NC-DC; NC-DN
P32942	Intercellular adhesion molecule 3 (ICAM3)	6.9678	0.001568	2.8045	0.024	DN-DC; DN-NC	DN-DC; NC-DN

**Table 2 proteomes-12-00029-t002:** Parameters obtained with ROC analysis with the six identified proteins to distinguish between the three groups of patients in terms of sensitivity and specificity.

Parameter	NC vs. DN	DC vs. DN	DC vs. NC
AUC	0.829	0.840	0.876
*p* ^†^	0.0015	0.0033	<0.0001
Sensitivity, %	90	77	79
Specificity, %	70	90	87
PPV, %	74	88	85
NPV, %	88	79	81
FDR, %	26	12	15

† Model significance, chi-square test. AUC: Area under the curve; Sensitivity: [True positives/(True positives + False negatives)]; Specificity: [True negatives/(True negatives + False Positives)]; PPV: positive predictive value or precision, calculated as [True positives/(True positives + False positives)], or (1 − FDR); NPV: negative predictive value, calculated as [True negatives/(True negatives + False negatives)]; FDR: False discovery rate, equivalent to [False positives/(False positives + True positives)].

## Data Availability

The datasets generated and/or analyzed in the current study are available from the corresponding author upon reasonable request.

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
