# Peer review of "Circulating Factors as Potential Biomarkers of Cardiovascular Damage Progression Associated with Type 2 Diabetes"

_proteomes, 2024, doi:10.3390/proteomes12040029_

Round 1

Reviewer 1 Report

Comments and Suggestions for Authors

Dear authors, I read your manuscript carefully, and find it interesting. It is a logical continuation of your previous work. As you stated, the number of samples is low, and it needs more work to assign the identify proteins as true biomarkers. Also, the next step must include a more balanced amount of males and females subjects, and it must include healthy subjects. To improve the current manuscript, the results could describe the meaning of the plots, in its current format the complete meaning is restricted to some specialized researches, then the relevance of the work will be less appreciated. Finally for the discussion, the meaning, and the importance of the discover will be better understand including one or two figures showing the mentioned signaling pathways, and the possible interactions among the identified proteins.   

Author Response

Comments and Suggestions for Authors

Dear authors, I read your manuscript carefully, and find it interesting. It is a logical continuation of your previous work. As you stated, the number of samples is low, and it needs more work to assign the identify proteins as true biomarkers. Also, the next step must include a more balanced amount of males and females subjects, and it must include healthy subjects. To improve the current manuscript, the results could describe the meaning of the plots, in its current format the complete meaning is restricted to some specialized researches, then the relevance of the work will be less appreciated. Finally for the discussion, the meaning, and the importance of the discover will be better understand including one or two figures showing the mentioned signaling pathways, and the possible interactions among the identified proteins.

Response: We thank the Reviewer for giving us the opportunity to improve our paper. We modified the text, with all the changes made highlighted in yellow color, taking into account the suggestions and recommendations.

Regarding the plots, additional information has been added in the legends of the figures. However, the explanation of the specific background linked to PCA analysis goes beyond the scope of the paper, and readers are suggested to refer to specific literature, such are quoted reference no. 13. In order to make better evidence of the interrelationships between the identified proteins, also following the suggestion of Reviewer no. 2, the figure with all the pairwise linear correlations has been now presented in the manuscript.

For the Discussion section, as suggested by the Reviewer, we added an additional explaining sketch (Figure 5).

Reviewer 2 Report

Comments and Suggestions for Authors

The study by Sartore et al. aimed to identify potential biomarkers of coronary heart disease (CHD) in patients with type 2 diabetes (T2D) using a targeted proteomic approach based on proximity extension assay (PEA). The study has an interesting design, comparing three groups: T2D patients without cardiovascular disease (CVD), T2D patients with CVD, and patients with CVD but without diabetes. However, as mentioned by the authors in the introduction, this study is closely related to a recent publication by the same group (doi: 10.1186/s12933-024-02125-1), which investigated biomarkers in a similar population using an untargeted approach based on liquid chromatography and mass spectrometry (LC-MS). The differential protein profiles identified in the two studies are distinct, with no common proteins found between them. It would be valuable to explore whether there is any relationship between the differential proteins identified in the current study and those from the previous study. A detailed comparative discussion of both studies is necessary to better understand the novelty of the current study and the potential pathophysiological relevance of the results.

The authors acknowledge the study's limitations, such as the small sample size and the absence of a control group, and have addressed these in the discussion section. However, to avoid potential misunderstandings, the discussion should be revised. Statements like "This association appears to be causal..." (lines 237-238) should be avoided, and comments on the potential role of the differential proteins in relation to diabetes are somewhat speculative. The authors should consider toning down and possibly shortening the discussion section.

Other major comments

- ROC Analysis (Fig 3) is a relevant result supporting the study results. AUCs for each protein should be provided to better understand their contribution to the provided curves. Are the AUCs for the 3 ROC curves in Figure 3 statistically significant?

- The authors separate the 6 differential proteins into two clusters based on the expression profile: a- PRCP, CES1, C2, ICAM3 and b- REG1A and IGLC2. It would be useful to include the FC in table 1.

Minor comments

-     -   The vertical axe legend is missing in figure 1 e.g. expression log2(NPX)

-      -  Figure 2 shows a biplot, (scales of loading axis are missing).

-     -   The horizontal axis in figure S1 is shifted to the right in both panels. The legend to figure S1 indicates that the following are plotted “distribution of Normalized Protein expression (NPX) in the overall samples”. There is no table with the normalized results in each group for the 92 proteins.

-       -  Adding the linear correlation shown in figure S4 would improve the understanding of the interpretation of the results.

-       -  In the discussion, from lane 199 to 202, the reference for PRCP function is missing.

Author Response

Comments and Suggestions for Authors

The study by Sartore et al. aimed to identify potential biomarkers of coronary heart disease (CHD) in patients with type 2 diabetes (T2D) using a targeted proteomic approach based on proximity extension assay (PEA). The study has an interesting design, comparing three groups: T2D patients without cardiovascular disease (CVD), T2D patients with CVD, and patients with CVD but without diabetes. However, as mentioned by the authors in the introduction, this study is closely related to a recent publication by the same group (doi: 10.1186/s12933-024-02125-1), which investigated biomarkers in a similar population using an untargeted approach based on liquid chromatography and mass spectrometry (LC-MS). The differential protein profiles identified in the two studies are distinct, with no common proteins found between them. It would be valuable to explore whether there is any relationship between the differential proteins identified in the current study and those from the previous study. A detailed comparative discussion of both studies is necessary to better understand the novelty of the current study and the potential pathophysiological relevance of the results.

Response: We express our gratitude to the Reviewer for careful evaluation of the manuscript and for words of appreciation on the work conducted. We modified the manuscript text, highlighted in yellow color, taking into account all the suggestions and recommendations.

We thank the Reviewer also for the suggestion to make a comparison between the proteins found in the present study and those identified in our previous investigation. Therefore we compared the profile of the proteins from both studies by means of a multivariate approach (PCA) and found that the overall behaviour of the protein pattern seems to indicate, besides absence of any overlapping, a complementary profile of the proteome, so extending the candidates of potentially useful markers for cardiovascular diabetes complications. A new PCA graph has been added as supplementary Figure S6, and discussion provided.

The authors acknowledge the study's limitations, such as the small sample size and the absence of a control group, and have addressed these in the discussion section. However, to avoid potential misunderstandings, the discussion should be revised. Statements like "This association appears to be causal..." (lines 237-238) should be avoided, and comments on the potential role of the differential proteins in relation to diabetes are somewhat speculative. The authors should consider toning down and possibly shortening the discussion section.

Response: We thank the Reviewer for suggestions about discussion section, which now has been revised.  Following also the suggestion of Reviewer no.1, we added a new figure with a scheme of the possible role of the identified proteins in the pathogenetic process of diabetes cardiovascular complications.

Other major comments

- ROC Analysis (Fig 3) is a relevant result supporting the study results. AUCs for each protein should be provided to better understand their contribution to the provided curves. Are the AUCs for the 3 ROC curves in Figure 3 statistically significant?

Response: We thank the Reviewer for this suggestion. A new table with the complete parameters of ROC analysis for each single protein have been presented in a new Table (Table S3), and mentioned in Results section. Also the significance of the AUC has been now inserted in Table 2.

- The authors separate the 6 differential proteins into two clusters based on the expression profile: a- PRCP, CES1, C2, ICAM3 and b- REG1A and IGLC2. It would be useful to include the FC in table 1.

Response: Due to the fact that 3 groups are considered, the method of FC was not performed, since it is suitable for two-group univariate data analysis method. However, the p-value, here obtained with ANOVA and post-hoc tests, already takes into account the fold-change, in order to show proteins that are present at statistically significant level. To complete the Table, we added further information in Table 1 by adding a column with the False discovery rate (FDR) for each protein.

Minor comments

-     -   The vertical axe legend is missing in figure 1 e.g. expression log2(NPX)

Response: The y-axis legend has been added to the figure.

-     Figure 2 shows a biplot, (scales of loading axis are missing).

Response: The loading axes have been added to the figure, and explained in the legend.

-     The horizontal axis in figure S1 is shifted to the right in both panels. The legend to figure S1 indicates that the following are plotted “distribution of Normalized Protein expression (NPX) in the overall samples”. There is no table with the normalized results in each group for the 92 proteins.

Response: The x-axis labels of figure S1 have been aligned.  A new table (Table S3) has been added with NPX data for each group of patients.

-       Adding the linear correlation shown in figure S4 would improve the understanding of the interpretation of the results.

Response: As suggested, the graph with multiple correlations has been inserted in the manuscript as Figure 3. The statistical details have been reported in Table S4.

-       In the discussion, from lane 199 to 202, the reference for PRCP function is missing.

Response: Quotation to reference has been indicated.